# Microfluidic Technology for Measuring Mechanical Properties of Single Cells and Its Application

**DOI:** 10.3390/bioengineering11121266

**Published:** 2024-12-13

**Authors:** Yixin Yin, Ziyuan Liu

**Affiliations:** Beijing Key Laboratory of Restoration of Damaged Ocular Nerve, Department of Ophthalmology, Peking University Third Hospital, Peking University Institute of Laser Medicine, No. 49 North Garden Road, Haidian District, Beijing 100191, China; 2110301236@stu.pku.edu.cn

**Keywords:** single-cell mechanics, cell mechanical properties, microfluidics

## Abstract

Cellular mechanical properties are critical for tissue and organ homeostasis, which are associated with many diseases and are very promising non-labeled biomarkers. Over the past two decades, many research tools based on microfluidic methods have been developed to measure the biophysical properties of single cells; however, it has still not been possible to develop a technique that allows for high-throughput, easy-to-operate and precise measurements of single-cell biophysical properties. In this paper, we review the emerging technologies implemented based on microfluidic approaches for characterizing the mechanical properties of single cells and discuss the methodological principles, advantages, limitations, and applications of various technologies.

## 1. Introduction

Cells in vivo are continuously subjected to a variety of mechanical forces in the cellular microenvironment, such as shear, compressive, and tensile forces. These mechanical forces directly affect the mechanical properties of cells [1]. Cellular mechanical properties are critical for tissue and organ homeostasis [2], which are associated with many diseases Cellular mechanical properties are defined as the ability or resistance of a cell to deform when subjected to external mechanical forces. The study of cell mechanics focuses precisely on exploring the response of cells to these mechanical forces and how this response can reveal the biological properties and functions of cells in physiological or pathological states. In recent years, significant advances have been made in tools to measure the mechanical properties of cells, and these properties have been found to be strongly associated with many human diseases, such as metastatic cancers, inflammation, and blood disorders [3,4,5,6,7]. The cellular mechanical properties of cells have been characterized by a wide range of changes in the cellular mechanics. Such changes not only reflect the physiological state of the cell but may also be a key marker for early diagnosis and disease prognosis.

Cell separation usually relies on biochemical and biophysical markers. Although biochemical markers have a broad base of applications, biophysical markers, such as the mechanical properties of cells, still show unique advantages. They serve as non-labeled biomarkers, avoiding expensive markers and complex sample preparation processes while enabling contamination-free and contact-free separations, which are important in clinical and laboratory applications.

Microfluidics, an advanced technique for manipulating and studying fluid behavior at the submillimeter scale [8], is considered to be one of the most promising current methods for studying single-cell mechanics [9,10,11]. It exploits the mechanical properties of cells—including deformability, viscoelasticity, stiffness, size, and shape—for cell separation and mechanical characterization. This technique not only improves the accuracy of single-cell analysis but also extends the range of applications of cell mechanics research. In this review, we provide a systematic summary of existing microfluidic techniques, focusing on the characterization of single-cell mechanical properties. We categorize the techniques according to the way of inducing cell deformation (Figure 1, Table 1) and describe in detail the methodology, principle, and working mechanism of each technique. In addition, we explore the specific application scenarios of these techniques, analyze their advantages and limitations, and propose possible directions for clinical applications with a view to providing references for future research and practice.

## 2. Electroporation-Induced Deformation

Electroporation techniques disrupt the integrity of the cell membrane by applying an electric field of high voltage outside the cell. Due to the relatively weak hydrophobic/hydrophilic interactions of the cell membrane’s phospholipid bilayer, this electric field creates temporary holes in the phospholipid bilayer of the cell membrane [23,24,25]. This process can be divided into two steps: First, water molecules penetrate the bilayer to form an unstable hydrophobic pore, and then the polar head groups of adjacent lipids are redirected toward these water molecules to form a stable hydrophilic pore, allowing polar molecules to enter the cell [26]. After electroporation, the cell membrane can repair itself to some extent, allowing the cell to remain intact [25].

Cancer cells undergo significant changes in their mechanical properties due to alterations in the cytoskeleton. Two consistent findings have been observed across multiple measurement platforms: cancer cells are more deformable than normal cells, and cancer cell deformability is associated with an increased likelihood of metastasis [2]. Bao N et al. [2] hypothesized that cell deformability is related to how much a cell swells during electroporation. They used microfluidic electroporative flow cytometry (EFC) to determine the swelling of single cells at a fluence of ~5 cells/s. EFC combines microfluidic technology and electroporation to enable the detection of cellular deformations through the following steps (Figure 2):i.Electric field application: a constant DC voltage is applied to the microchannel so that the cells undergo electroporation in response to the electric field.ii.Cell swelling: electroporation temporarily disrupts the cell membrane, allowing water and small molecules to enter the cell, resulting in an increase in cell size.iii.Recording and analysis: a CCD camera is used to record the passage of cells through the microfluidic channel and to observe and quantify the swelling of single cells. The swelling data can be used as an indicator of cell deformability.

As a result, it was found that cancer cells have a higher deformation ability compared to normal cells. This deformability was directly correlated with the metastatic ability of the cells. Specifically, non-cancerous breast epithelial cells MCF-10A had lower deformability, non-metastatic human breast epithelial cancer cells MCF-7 had increased deformability compared to MCF-10A, and metastatic breast cancer cells (TPA-treated MCF-7) exhibited significantly higher deformability. This suggests that cell deformability correlates with cancer invasiveness and metastasis and may serve as a label-free, quantitative assay for cancer diagnosis and staging. The deficiency is the lack of relevant evidence that the degree of cell swelling is related to the ability of cells to deform because the degree of cell swelling may also be related to cell size and membrane stability [27].

## 3. Hydrodynamic Stretching-Induced Deformation

One of the major problems with early microfluidic technologies that measured the mechanical properties of single cells was the low throughput (~5 cells/s), which was orders of magnitude different than traditional flow cytometry. This low throughput limited the detection of large cell populations and hindered the generation of statistically significant data.

An automated microfluidic technique, deformational cytometry (DC), greatly improves this problem by combining inertial focusing, hydrodynamic stretching, and automated image analysis, allowing single-cell deformations to be detected at a rate of approximately 2000 cells/s [4]. Inertial focusing is a sheathless method for arranging and measuring flowing cells [28]. This method abandons the traditional solid material microfluidic channel wall, allowing the cells to be carried and measured around the fluid, avoiding contact, adhesion, or fouling of the channel surface to avoid blockage. In addition, this method produces high strains, is easy to observe, and has a high strain rate, providing unique cellular mechanical insights.

Cells are transported at high speed in a narrow flow line near the center of the microfluidic channel to the junction of two cross channels, where they change the direction of travel and enter the stretch flow, where they are mechanically stretched and deformed (Figure 3). At the same time, high-speed cameras capture images of cell deformation, and automatic image analysis algorithms quantify cell size and degree of deformation and create a 2D scatter plot.

The results show that cancer cells in pleural fluid are larger and more easily deformed than benign cells. This method achieves both high test speeds and greater deformation of the cells under test. The authors used this method to detect malignant cells and leukocytes in pleural fluid and accurately predicted the disease state of cancer patients with 91% sensitivity and 86% specificity. This suggests that this technique not only improves detection throughput but also provides an effective complement to histologic analysis and biopsy of pleural effusions after thoracentesis, making up for the shortcomings of traditional manual pathology biopsies.

Since then, they have reported an improved method [3], in which inertially focused cells are squeezed vertically at high speed in the flow by a cell-free fluid extracted from the cell suspension, increasing the throughput by more than an order of magnitude (65,000 cells/s). This method is improved by eliminating the setup of the stretching flow so that the cells always maintain the original direction of travel, thus avoiding the limitation of the time (tens of microseconds) that the cells stay at the stagnation point of the stretching flow and increasing the detection efficiency.

Otto et al. [12] proposed real-time deformational cytometry (RT-DC), a microfluidic approach to flow cytometry based on the principle of constriction channeling, which can analyze large cell populations (>100,000 cells) at rates greater than 100 cells/s. RT-DC can analyze cell cycle phases, track stem cell differentiation, and identify different cell populations in whole blood by mechanical fingerprinting. This method determines the viscoelasticity of erythrocytes, granulocytes, and peripheral blood mononuclear cells in a single measurement and enables differentiation of B and CD4+ T lymphocytes by measuring mechanical properties [29]. Tavassoli et al. [17] also used this method for label-free isolation and biophysical characterization of cardiomyocytes, achieving rapid (<2 h for the entire process), high-throughput (120,000 cells/min), and high-purity cell isolation. Studies have shown that this method can be used to predict the disease status of inflammation and malignant tumors quickly and without bias after surgery and can obtain the single-cell physical characteristics of surgical biopsy tissue within half an hour and detect its malignancy, eliminating the freezing and staining steps of traditional biopsy methods [30].

## 4. Micropipette Aspiration-Induced Deformations

A micropipette is a precision instrument for handling tiny volumes of liquid. The principle of operation is to aspirate liquids or cells by generating negative pressure at the tip of the pipette (Figure 4). In cell mechanics studies, the magnitude of this negative pressure can be used to characterize the deformability of the cell, i.e., how much the cell deforms when a force is applied. This method has been widely used in the study of the mechanical properties of normal and abnormal erythrocytes [31,32,33]. One of its major advantages over other methods of studying cell mechanics is that it is simple and straightforward to operate, but it suffers from the problems of bulky equipment and low throughput.

To overcome the shortcomings of traditional micropipettes, researchers have combined microfluidic technology with micropipettes, a combination that can significantly increase the throughput of the process. Kim et al. [34] achieved efficient separation of plasma from whole blood by using this combination of microfluidic pipetting and microfluidic technology. Their method has very high purity (99.88%) and throughput (904.3 μL/min). This technique is particularly suitable for pre-transfusion plasma preparation because it is fast, economical, and does not require complex equipment.

This method has also been utilized in other cell types. By this method, Young’s modulus was rapidly measured in human breast non-tumorigenic (MCF-10A) and metastatic cancer (MDA-MB-231) cells [13]. This measurement helps to distinguish between these two different types of cells, thus providing valuable data for cell mechanics studies.

The combination of applying microfluidic technology to micropipettes not only improves the efficiency of the operation but also expands its potential for application in the field of cell mechanics. This technological advancement provides new solutions for cell separation, allowing related research to be conducted with greater efficiency and precision.

## 5. Constriction Channel-Induced Deformation

The diameter of the constriction channel is slightly smaller than the diameter of the test cell, and as the cell passes through the channel, it is deformed by pressure from the channel wall. Parameters related to cell deformability, such as transmission time, can be quantified to indirectly characterize cell deformability. Deformability is essential for the passage of cells through tiny channels such as capillaries. The high deformability of erythrocytes allows them to pass smoothly through capillaries smaller than their diameter, which is necessary for the normal functioning of the macro- and microcirculation. The constriction channels can simulate the environment of capillaries in the body. With the application of high-speed imaging or electrical impedance measurement, a higher throughput can be achieved compared to most other deformability measurement methods. Owing to these advantages, the constriction channels have been employed to measure the deformability of red blood cells [7,15,35,36], white blood cells [37], and cancer cells [16,38,39].

Rosenbluth et al. [6] simulated capillaries with microchannels of different diameters and measured cell transport times with a high-speed camera (Figure 5). They explored changes in blood cell deformability in disease models, laying the groundwork for clinical applications of the technique. They used the technique to study changes in the deformability of healthy versus malaria parasite-infected erythrocytes at different stages of the disease. The results showed that malaria parasite infection decreased the deformability of erythrocytes. In addition, they studied the deformability of leukocytes in inflammatory states, demonstrating the clinical relevance of the device in sepsis and leukocyte stasis. Their measurements were able to clearly show the differences between the cells of patients with symptomatic and asymptomatic leukosis.

Lange et al. [14] quantified the mechanical properties of suspension leukemia cell K562 and mouse embryonic fibroblast cell line NIH 3t3 cells by means of an array containing eight micro-contractile channels and analyzed the relationship between protein expression and cellular mechanical properties in conjunction with fluorescence imaging, confirming that the modulus of cellular elasticity rises with increasing levels of laminin A while cell mobility decreases. To reduce the variability between measurements, they used a histogram matching of pressure, strain, and protein expression levels, which selects and analyzes only those cells that have experienced the same pressure and strain, effectively improving the repeatability of measurements.

The large amount of data generated by high-speed cameras (several gigabytes per second) requires significant computational resources and time to process. For this reason, some studies have turned to more efficient data acquisition through electrical impedance measurements, where the transmission time, the amplitude ratio, and the phase difference between with cells and without cells are obtained as the erythrocytes flow through the constricted channel [40,41]. Zheng et al. [15] used this method to efficiently measure cell biophysical properties in large samples at 100–150 cells per second. The success rate of this method in classifying fetal/neonatal and adult red blood cells was confirmed by backpropagation neural networks. This measurement is purely electrical and does not require microscopic imaging, thus potentially enabling more efficient measurements. Moreover, because almost all electric field lines penetrate the RBC’s membrane and the hemoglobin inside, the device is more sensitive to small biophysical differences than previous methods. They then used the method to measure the deformability of activated lymphocytes from patients with chronic lymphocytic leukemia and showed that the deformability of lymphocytes from patients with slow gonorrhea was lower than that of controls [37]. Regarding the measurements and detection of the approach of the cell to the sensor surface, there are many methods of testing, such as cell-based field-effect transistors (FET), which are semiconductor devices that use an electric field to regulate the flow of current, can sense small changes in charge, and can convert these charge changes into measurable electrical signals, so as to achieve sensing functions, which can be used to detect changes in cell membrane potential, and then monitor cell activity and signal conduction [42,43].

The transmission time is affected not only by cell deformability but also by cell size and its friction properties with the channel wall. In view of this, Byun et al. [16] developed a suspended microchannel resonator (SMR) with an integrated constriction device, which allowed for the combined measurement of the buoyant mass and transport time of cells in a microfluidic channel and was used to efficiently differentiate between different types of lung adenocarcinoma cells, including metastatic and non-metastatic cells. Further, using SMR, they tracked the rate at which cells flow through the contractile channel and inferred the relative importance between the deformability of cancer cells and surface friction.

## 6. Optical Stretching-Induced Deformation

Optical stretching utilizes two back-propagating unfocused laser beams to exert an action on a dielectric object. These two laser beams create two orthogonal induced forces, the gradient force and the axial force, with the gradient force pulling the cell toward the fiber centerline [44]. The axial force pushes the cell away from the ends of the fiber, stretching the cell in the direction of the beam axis (Figure 6). Guck et al. [18] utilized this principle to develop the first optical stretching device in 2000, which can be used to measure the dielectric viscoelasticity of materials, including cells. Through a combination of optical stretching and microfluidics, i.e., the use of optical fibers to capture and stretch flowing cells, they measured the optical deformability of the mouse fibroblast cell line BALB/3T3 and its cancerous counterpart and human mammary epithelial cells and their cancerous counterparts and found that the optical deformability of highly metastatic breast cancer cells, MDA-MB-231, was significantly reduced by all-trans retinoic acid treatment, which was attributed to the fact that all-trans retinoic acid reduced the metastatic properties of this cell [5]. Huang et al. [19] developed a microfluidic chip combining optical fibers and electrodes to measure the mechanical and dielectric properties of normal cells (e.g., HepaRG, MCF10A) and cancer cells (e.g., HeLa, A549, and MCF7), achieving an integrated measurement of multi-physical field parameters and further validating the potential of this method for cancer cell analysis.

The main advantage of optical stretching is that it measures the mechanical properties of the cells without direct contact with the cells, and no radiation damage to the cells was observed throughout the process. This non-destructive and non-contaminating technique may have important applications in stem cell therapy.

## 7. Acoustic Wave-Induced Deformation

When sound waves are reflected, refracted, scattered, and other effects occur on the surface of an object, there is an exchange of momentum and energy with the object. This exchange is manifested macroscopically as the sound wave exerts a force on the object, called the acoustic radiant force (ARF) (Figure 7). Using ARF or by quantifying the resulting displacement, cell separation and mechanical characterization can be achieved [45,46]. Many studies have shown that acoustic-based cell sorting has great potential for maintaining cell viability, high purity, and high recovery [47,48,49,50,51].

Kang et al. [20] used vibrations from SMR as an acoustic energy source to quantify the resonant frequency shift of the cell at the point where the vibration amplitude is a zero minimum and confirmed that this shift correlates with the acoustic scattering at the cell surface, which, when the cell size is normalized, depends on the mechanical properties of the cell. Their method enables continuous, non-invasive measurement of the mechanical properties of single cells at high temporal resolution over long periods of time, which is crucial for the observation of small changes in cell transients. The authors used this method to characterize the dynamics of cellular mechanical properties during mitosis, providing a basis for understanding how growing cells maintain mechanical integrity and demonstrating that acoustic scattering can be used to non-invasively probe fine and transient dynamics.

Li et al. [21] used a radiofrequency (RF) generator to produce focused traveling surface acoustic waves (FTSAW) [52,53,54] to deflect target cells to the target outlet for collection. They achieved high throughput (>100 cells/s) and high purity (~91.8%) cell sorting to separate live cells from a mixture of fixed and live MCF-7 cells.

Romanov et al. [22] measured the viscoelasticity of human embryonic kidney cells HEK293T using the acoustic force spectroscopy (AFS) technique. The AFS technique calculates the stiffness of the cells by driving tens to hundreds of silica beads attached to the cells away from the cell surface by acoustic waves and calculating the stiffness of the cells by measuring the mean-square displacements (MSDs) of the silica beads under Brownian motion and determining the diffusion coefficients using the Einstein relation. One of the major advantages of the AFS technique is the “plug-and-play” format: the cells to be analyzed are simply loaded into a microfluidic device (AFS chip), and the chip is connected to a mobile platform. The “plug-and-play” format: the cells to be analyzed are simply loaded into a microfluidic device (AFS chip), and the chip is connected to a mobile platform to complete the measurement. This design makes force calibration and viscoelasticity measurement more intuitive and easier.

## 8. Conclusions

In this review, we systematically summarize six methods for inducing cell deformation in different ways to measure the mechanical properties of single cells. We describe in detail the basic principles and operational mechanisms of each method, providing insights into their unique features and their respective advantages. For example, some methods may accurately measure the mechanical response of cells by applying microfluidics to apply precise pressure or shear stresses, while others may utilize microstructural arrays to enable high-throughput determination of cellular mechanical properties. In addition, we explore the wide range of applications of these technologies in several fields, including cell separation techniques, disease diagnosis, and analysis of pathophysiological states, where the needs of these fields drive the continuous advancement of mechanical measurement technologies.

Our discussion focuses not only on the use of these methods in laboratory research but also addresses their potential for practical clinical applications. For example, cellular mechanical properties can be used as an important indicator for early screening of tumors or for assessing disease progression and efficacy. These applications demonstrate the important role of single-cell mechanical measurement techniques in advancing biomedical research and clinical practice.

Going forward, we anticipate that the measurement of single-cell mechanical phenotypes will become more accurate and comprehensive as the technology continues to improve in terms of throughput, degree of automation, and integration. Advances in microfluidics will further advance the field, enabling us to realize efficient and reliable cellular mechanical measurements on a larger scale. Over the past two decades, microfluidic-based single-cell mechanical characterization has progressed significantly, with orders of magnitude improvements in test throughput, simpler procedures, and significant improvements in the sensitivity and specificity of results. These advances not only enhance the relevance of clinical applications but also provide a solid foundation for future technology development.

We have reason to believe that in the near future, with the continuous innovation and optimization of the technology, there will be higher throughput and more robust and comprehensive techniques for measuring the mechanical properties of cells. These technologies are expected to be more widely used in clinical applications, thus playing a greater role in personalized medicine, early disease diagnosis, and prognosis assessment.

## Figures and Tables

**Figure 1 bioengineering-11-01266-f001:**
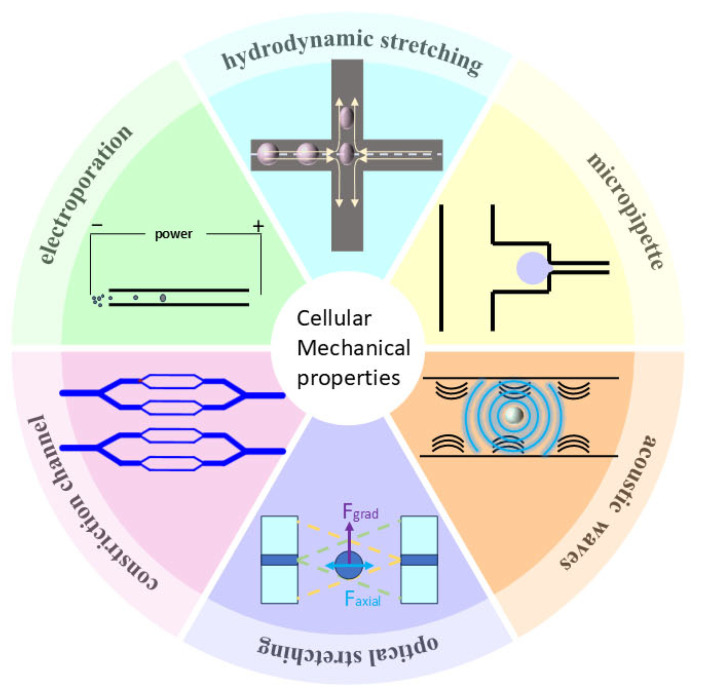
Classification of microfluidic techniques for mechanical characterization of single cells.

**Figure 2 bioengineering-11-01266-f002:**
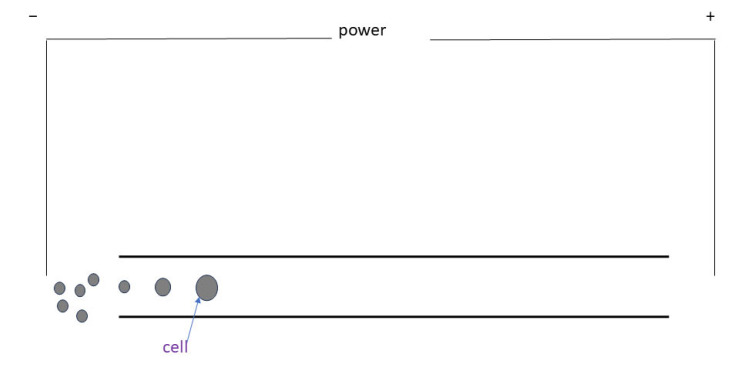
Schematic diagram of EFC.

**Figure 3 bioengineering-11-01266-f003:**
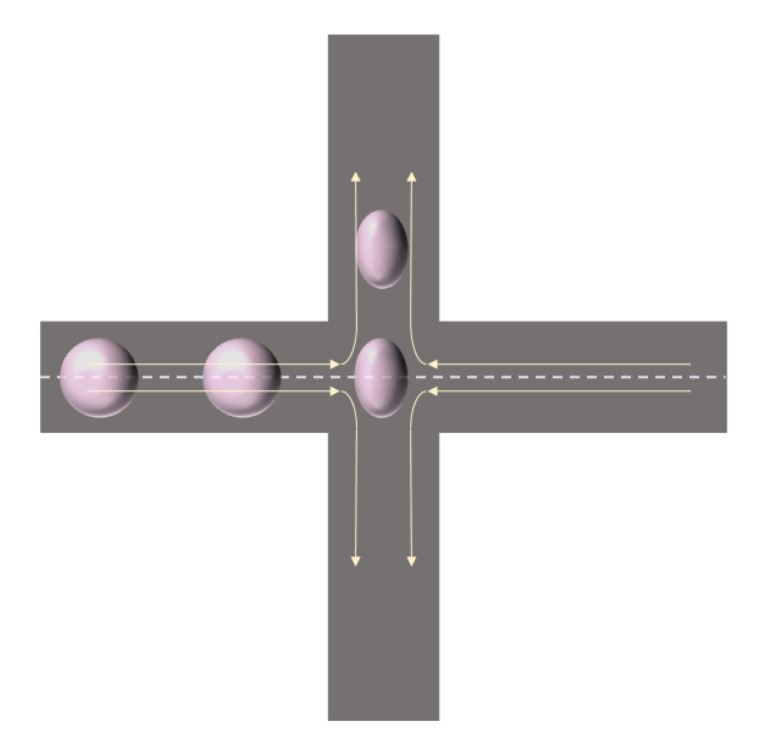
Schematic diagram of the hydrodynamic stretching-induced deformation.

**Figure 4 bioengineering-11-01266-f004:**
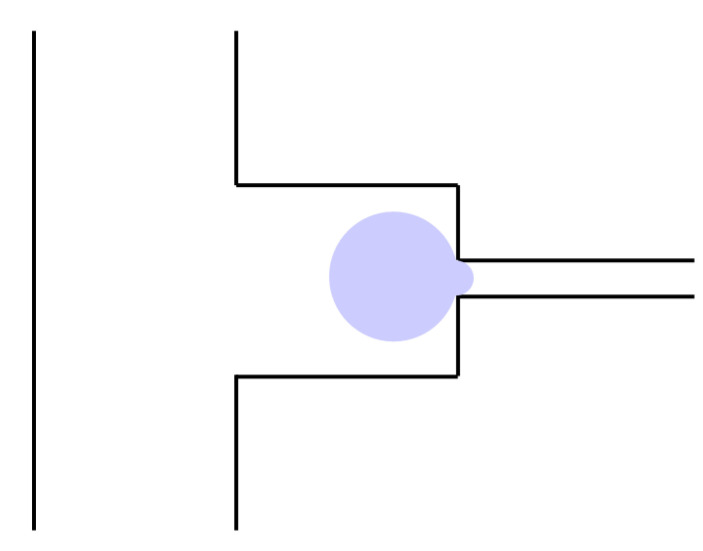
Schematic diagram of micropipette.

**Figure 5 bioengineering-11-01266-f005:**
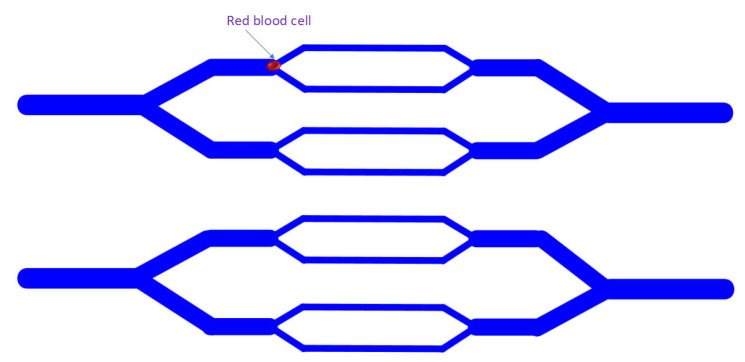
Schematic diagram of constriction channel.

**Figure 6 bioengineering-11-01266-f006:**
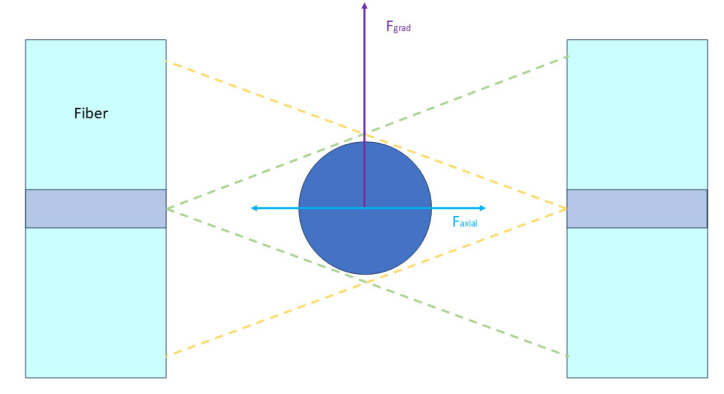
Schematic diagram of optical stretching.

**Figure 7 bioengineering-11-01266-f007:**
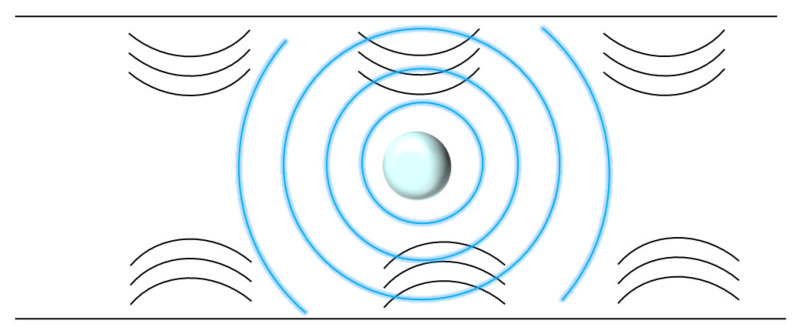
Schematic diagram of the acoustic wave-induced deformation.

**Table 1 bioengineering-11-01266-t001:** Microfluidic technologies for single-cell mechanical characterization.

Methods	Authors and References	Targeted Cells	Measured Quantity	Characterized Physical Properties	Advantages
Electroporation	Bao et al. [3]	MCF-7, TPA-treated MCF-7, MCF-10A	cell swelling degree	deformability	laying a foundation for measuring cell mechanical properties by electroporation technique
Hydrodynamic Stretching	Gossett, D.R. et al. [5]	malignant cells and leukocytes in pleural fluid	cell diameter and the aspect ratio of the maximally deformed cell	deformability	allowing cells to be measured around the fluid to avoid contact, adhesion, and blockage of the channel surface
	Dudani, J.S. et al. [4]	HeLa	cell diameter and the aspect ratio of the maximally deformed cell	deformability	increasing the throughput by more than an order of magnitude (65,000 cells/s)
	Otto et al. [12]	blood cell	circularity and cross-sectional area of single cell	Deformability, Young’s modulus	real time, analyzing large cell populations (>100,000 cells) at rates greater than 100 cells/s
Micropipette	Lee et al. [13]	MCF-10A, MDA-MB-231	the protrusion length of the trailing edge of the cell into the pipette, the radius of the cell outside the pipette, pipette radius	Young’s modulus	providing an alternative to cell migration assays that take minutes instead of hours
Constriction Channel	Rosenbluth et al. [7]	healthy erythrocytes, malaria parasite-infected erythrocytes	cell transit time through microfluidic channels	deformability	using microchannels to simulate capillaries, which has strong clinical relevance
	Lange et al. [14]	K562, NIH 3t3	the pressure drop Δ*p* across the constriction, the radius of the undeformed cell, the constriction width, and the entry time (*t*_entry_) of the cells	Young’s modulus	using a histogram matching of pressure, strain, and protein expression levels reduces the variability between measurements
	Zheng et al. [15]	fetal/neonatal and adult red blood cells, lymphocytes of patients with chronic lymphocytic leukemia	impedance amplitude ratio, impedance phase increase, transit time	deformability	more efficient data acquisition through electrical impedance measurements
	Byun et al. [16]	lung adenocarcinoma cells	the buoyant mass, transit time, entry and transit velocities	deformability	the effect of cell size on the transit time was taken into account
	Tavassoli et al. [17]	cardiomyocytes	same as above	same as above	rapid (<2 h for the entire process), high-throughput (120,000 cells/min), and high-purity cell isolation
Optical Stretching	Guck et al. [18]	BALB/3T3 and its cancerous counterpart, human mammary epithelial cells, and their cancerous counterparts	the major and minor axes of the cell, aspect ratio, aspect force ratio	deformability, optical deformability	non-destructive, non-contaminating
	Huang et al. [19]	HeLa, A549, HepaRG, MCF7, and MCF10A	the time-varying diameter of the cell along the beam axis, and the original diameter of the cell along the same direction when the cell is trapped initially	shear modulus *G*, steady-state viscosity *ψ*, and relaxation time *τ*	achieving an integrated measurement of multi-physical field parameters by combining optical fibers and electrodes
Acoustic Waves	Kang et al. [20]	L1210, Baf3, S-HeLa	the size-normalized acoustic scattering (SNACS)	stiffness	continuous measurement, which can observe small changes in the cell
	Li et al. [21]	fixed and live MCF-7 cells	transit time	deformability	high throughput (>100 cells/s) and high purity (~91.8%) cell sorting
	Romanov et al. [22]	HEK293T	mean-square displacements	stiffness	“plug-and-play” format, more intuitive and easier

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
