# Peer review of "Microfluidic Technology for Measuring Mechanical Properties of Single Cells and Its Application"

_bioengineering, 2024, doi:10.3390/bioengineering11121266_

Round 1
Reviewer 1 Report
Comments and Suggestions for Authors
This review article addresses a highly promising and impactful research theme, focusing on microfluidic technologies for measuring the mechanical properties of single cells. The topic is of significant interest due to its potential for practical applications in biomedical research and clinical diagnostics. I believe that this review is suitable for publication, as it provides a comprehensive overview of the field and highlights critical developments.
However, I recommend the following revisions to enhance the clarity and overall value of the manuscript:
#1:Improve the Clarity of Figure 1:
The individual elements in Figure 1 are too small, making it difficult for readers to understand the content clearly. I suggest revising the figure to make each component larger and more legible, ensuring that the information is effectively conveyed.
#2:Create a Table Summarizing Citations:
To improve readability and organization, I recommend presenting the cited studies in a tabular format. This table could include details such as the referenced technology, its advantages, limitations, and applications. This would help readers quickly grasp the key contributions of each study and enhance the utility of the review
By addressing these points, the authors can further improve the accessibility and impact of this review article, making it an even more valuable resource for readers in the field.
Author Response
Comments 1: Improve the Clarity of Figure 1:
The individual elements in Figure 1 are too small, making it difficult for readers to understand the content clearly. I suggest revising the figure to make each component larger and more legible, ensuring that the information is effectively conveyed.
Response 1: Thank you for pointing this out. I have enlarged each of the small figures in Figure 1 and placed them in corresponding parts.
Comments 2: Create a Table Summarizing Citations:
To improve readability and organization, I recommend presenting the cited studies in a tabular format. This table could include details such as the referenced technology, its advantages, limitations, and applications. This would help readers quickly grasp the key contributions of each study and enhance the utility of the review
Response 2: I agree with this comment. I have made a table of the various technologies mentioned in this review and their target cells, detection indicators, characterized physical properties and advantages.
Reviewer 2 Report
Comments and Suggestions for Authors
This is an OK manuscript, but it is certainly not a review; it is rather a very short note on the subject of precise measurements of single-cell biophysical properties. The literature on the subject is huge and the authors are strongly encouraged to overview each part of it properly and in depth. A table e.g. should be presented with the info on which of the methods is beneficial for measuring which property and why. It should also be clearly formulated which property is actually measured by each method and with what precision, is it the ionic solution, cell-membrane properties, or something else. Regarding the measurements and detection of the approach of DNA and other charged entities such as cell membranes to the sensor surface, the authors can mention in the revised version the relevant Ref. [ https://doi.org/10.1016/j.bios.2013.02.026 ] as well as the related works in the field of field-effect-based detection for larger complexes, viruses, as well as for whole cells. After these modifications one more round of review is necessary prior to taking a final decision on this material.
Author Response
Comments 1: A table e.g. should be presented with the info on which of the methods is beneficial for measuring which property and why. It should also be clearly formulated which property is actually measured by each method and with what precision, is it the ionic solution, cell-membrane properties, or something else.
Response 1: Thank you for pointing this out. I have made a table of the various technologies mentioned in this review and their target cells, detection indicators, characterized physical properties and advantages.
Comments 2: Regarding the measurements and detection of the approach of DNA and other charged entities such as cell membranes to the sensor surface, the authors can mention in the revised version the relevant Ref. [ https://doi.org/10.1016/j.bios.2013.02.026IF: 10.7 Q1 ] as well as the related works in the field of field-effect-based detection for larger complexes, viruses, as well as for whole cells.
Response 2: Thank you for pointing this out. I have added references 19-22 to reinforce the explanation of the effect of electric fields on cell membranes.
Reviewer 3 Report
Comments and Suggestions for Authors
Dear authors
Your review on "Microfluidic Technology for Measuring Mechanical Properties of Single Cells and its Application" is interesting.
I would have some comments :
- one figure per method would allow a better understanding of the technique.
- You could put more examples of use of each method and add references because 24 references for a review is not enough
Best regards
Author Response
Comments 1: one figure per method would allow a better understanding of the technique.
Response 1: Thank you for pointing this out. I have given each method a corresponding picture.
Comments 2: You could put more examples of use of each method and add references because 24 references for a review is not enough.
Response 2: Thank you for pointing this out. I have increased the number of references to 35.
Round 2
Reviewer 2 Report
Comments and Suggestions for Authors
The authors conducted some revisions of the text, in particular the table added will be of value for the final review. Some of the methods of detection of cellular signals and of electric charges near the sensor surfaces were still not addressed in the revised text. The authors are thus encouraged to mention at least Refs. [https://www.sciencedirect.com/science/article/pii/S1084952109000184] and [https://link.springer.com/article/10.1007/s00216-013-6951-9]. Additionally, for a solid review i find the list of literature way too short: it must be extended substantially in order to provide the reader with an adequate and most complete overview of the recent relevant and important literature.
Author Response
Comments 1: Some of the methods of detection of cellular signals and of electric charges near the sensor surfaces were still not addressed in the revised text. The authors are thus encouraged to mention at least Refs. [https://www.sciencedirect.com/science/article/pii/S1084952109000184] and [https://link.springer.com/article/10.1007/s00216-013-6951-9].
Response 1: Thank you for pointing this out. I have added these two articles to the article as Reference 42,43.
Comments 2: Additionally, for a solid review i find the list of literature way too short: it must be extended substantially in order to provide the reader with an adequate and most complete overview of the recent relevant and important literature.
Response 2: Thank you for pointing this out. I have increased the number of references to 54.
Round 3
Reviewer 2 Report
Comments and Suggestions for Authors
A solid revision, the material can be accepted in the present form.